# Energy harvesting optical modulators with sub-attojoule per bit electrical energy consumption

M. de Cea [1], A. H. Atabaki[1] & R. J. Ram [1✉]

The light input to a semiconductor optical modulator can constitute an electrical energy supply through the photovoltaic effect, which is unexploited in conventional modulators. In this work, we leverage this effect to demonstrate a silicon modulator with sub-aJ/bit electrical energy consumption at sub-GHz speeds, relevant for massively parallel input/output systems such as neural interfaces. We use the parasitic photovoltaic current to self-charge the modulator and a single transistor to modulate the stored charge. This way, the electrical driver only needs to charge the nano-scale gate of the transistor, with attojoule-scale energy dissipation. We implement this 'photovoltaic modulator' in a monolithic CMOS platform. This work demonstrates how close integration and co-design of electronics and photonics offers a path to optical switching with as few as 500 injected electrons and electrical energy consumption as low as 20 zJ/bit, achieved only by recovering the absorbed optical energy that is wasted in conventional modulation.

[1] Research Laboratory of Electronics, Massachusetts Institute of Technology, Cambridge, MA, USA. ✉email: rajeev@mit.edu

Today, low-energy optical interconnection underlies the continued scaling in performance of information processing and communications for data-intensive applications, ranging from large-scale datacenters to high-performance computing[1–4]. Alongside these 'traditional' application spaces, new applications for energy-efficient optical links are emerging, including specialized interconnects for neural interfaces[5–7], ultrasound and magnetic resonance imaging (MRI)[8–10] or cryogenic readout[11] (Fig. 1).

In all these applications, the distance between the sensing site, where the signal of interest is acquired, and the main processor can range from centimeters to several meters. The conventional approach is to use electrical interconnects, which require power-hungry electrical amplification due to the weak nature of the sensed signals, and pose a challenge to scaling to massively parallel readout because electrical energy consumption at the sensing site is tightly constrained. For instance, a 2 K increase in brain temperature significantly affects its normal activity and can severely damage the tissue if sustained over a timescale of hours, limiting the total power dissipated in neural interfaces (Fig. 1b) to about 10 mW[12]. Likewise, the limited cooling power available in the lower temperature stages of cryostats (Fig. 1c) limits the power dissipation of the readout architecture to the 1–100 mW range depending on the operating temperature[13]. On a related issue, the amplification, sensing, readout, and delivery of electrical signals in MRI environments (Fig. 1d) is challenging due to the electromagnetic interference and induced heating caused by the high magnetic fields present in the chamber[10].

The use of photonic interconnects could offer significant advantages in terms of achievable interconnect density, power dissipation, and robustness to electromagnetic interference compared to conventional electronics-based solutions. The development of a compact, scalable, low-power opto-electronic transducer capable of converting the (weak) sensed electrical signal into a modulated optical signal with attojoule-scale electrical energy consumption and no need for amplification could enable optical readout of thousands of high-density, parallel interconnects through a sensing macro, schematically depicted in Fig. 1a. Since these applications are mostly targeted for electrophysiological sensing, only moderate data rates on the order of 10–100 MHz are necessary.

Compatibility and ease of integration with complementary CMOS technology is an additional, critical requirement to realize low-cost and large-scale systems for the consumer market. Optical modulators fabricated in a silicon platform are particularly attractive due to their maturity, scalability, and the possibility of monolithic integration with CMOS electronics. Numerous high-performance integrated silicon modulators have been reported in the literature (Fig. 2a, Supplementary Table 1), but to date, their electrical energy consumption has been limited to values near or above 1 fJ/bit, dominated by the need for large driving voltages (>500 mV$_{PP}$) due to the relatively weak electro-optic effect in silicon.

To overcome such limitations, efficient, high-speed modulators have been realized in a variety of alternative materials like lithium niobate[14] or organic polymers[15] (Fig. 2a, Supplementary Table 2). While improvements in terms of electrical energy dissipation are possible, typical values are still in the fJ/bit range. In addition, such approaches suffer either from limited scalability, high optical losses, or difficulty of integration with electronics.

Fundamentally, all modulators developed to date—on silicon and other materials—work by delivering external electrical charge

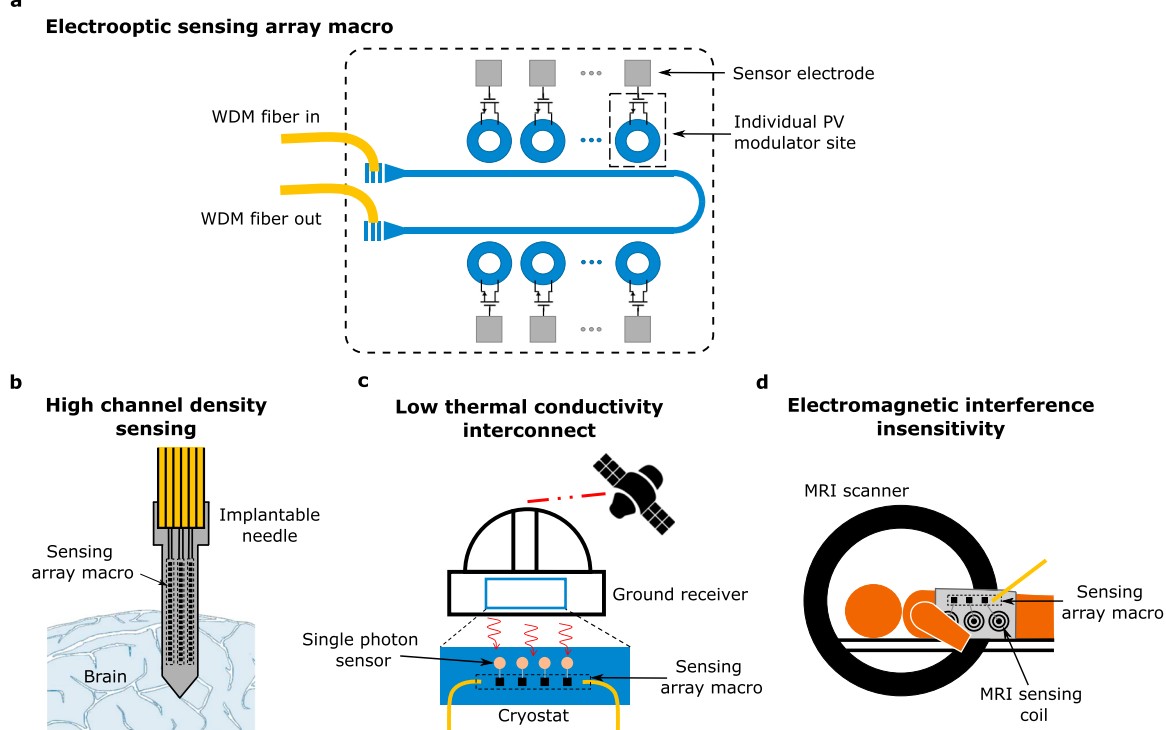

**Fig. 1 Efficient optical readout in sensing applications. a** Schematic of a low-power electro-optic transducer based on our photovoltaic (PV) modulator. Individual PV modulators connected to different sensing sites are addressed using different wavelengths through a wavelength division multiplexing (WDM) scheme. The development of such a compact, low-energy electro-optical transducer is essential for sensing applications where the electrical energy at the sensing site is constrained. Examples include massively parallel readout of electrophysiological signals (**b**), readout of single-photon sensors operating at cryogenic temperatures (**c**), and readout of electrical signals in environments with high electromagnetic interference, such as in magnetic resonance imaging (MRI) (**d**).

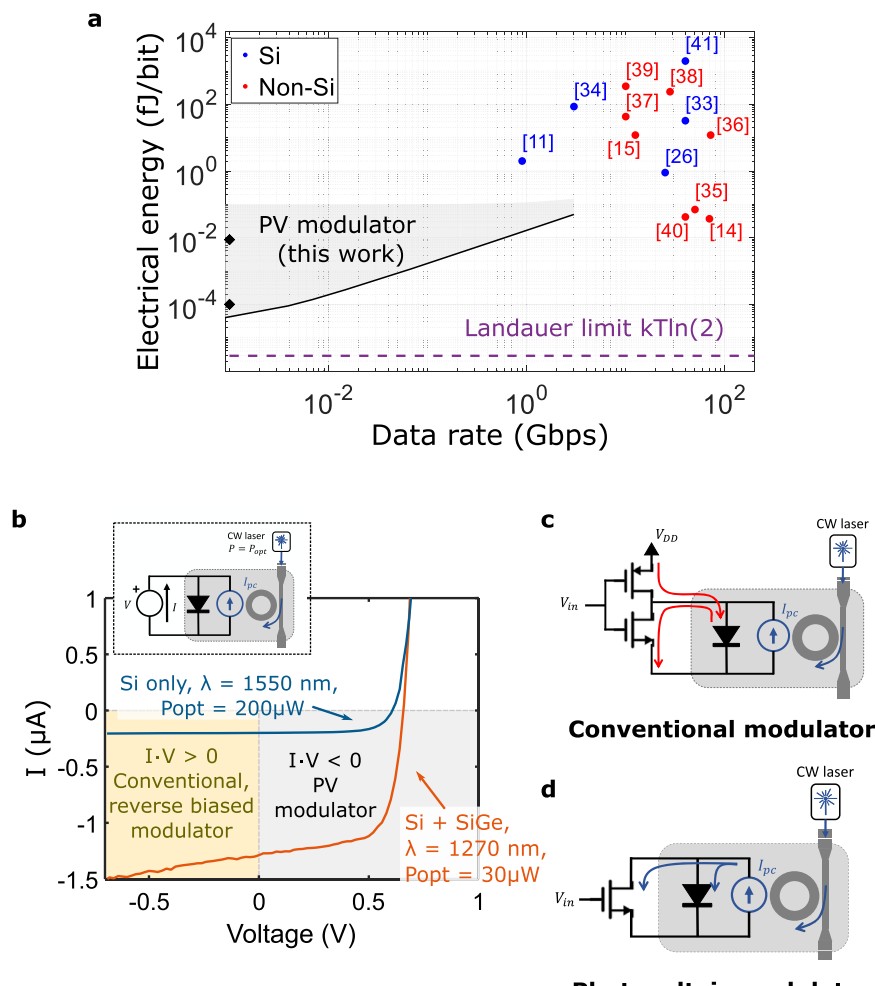

**Fig. 2 Photovoltaic modulation working principle. a** Electrical energy consumption and demonstrated data rate for the lowest power optical modulators reported in the literature[33-41]. Devices fabricated in silicon (blue) and other material platforms (red) are shown. Photovoltaic (PV) modulation (gray shaded area) can achieve orders of magnitude lower power consumption at speeds below ≈3 GHz, approaching the Landauer limit for binary switching (dashed purple line). Black diamonds show experimentally obtained optical modulation presented in this work. Supplementary Tables 1 and 2 show more information on the different devices. **b** Measured I–V curves of two different silicon photonic resonant modulators under illumination. Photocurrent generation is clear. Energy harvesting is possible when operating the device in the gray shaded area, while conventional modulators operate in the yellow shaded area. The inset shows the measurement setup. More information is given in Supplementary Discussion 2. **c** Conventional modulation driving scheme. The $V_{DD}$ source needs to provide the energy required to charge the modulator capacitance and that associated with the work on photogenerated charges. **d** PV modulator driving scheme. The photocurrent is used to self-charge the modulator capacitance and the transistor channel, greatly reducing the electrical energy provided by the driver.

to the electrodes of the device. On the other hand, photons are constantly delivered to the modulator, which generate free carriers through various sources of optical absorption—including interfacial and sub-bandgap absorption. These free carriers can be utilized for modulation, eliminating the need for supplying external charge to the modulator and its associated energy dissipation. While such potentially useful absorption can be enhanced by the addition of absorbing materials such as germanium (Ge), parasitic absorption mechanisms make photocurrent generation universal in semiconductor-based modulators. In modern silicon photonics devices—including Mach-Zehnder, microring, and microdisk geometries—this is dominated by surface state absorption[16-18] resulting in estimated responsivities close to 1 mA/W (Supplementary Discussion 1). We note that otherwise comprehensive reviews[19] have only considered the energy associated with these photocurrents in electro-absorption devices and neglected this contribution in electro-refractive devices.

Therefore, most, if not all, silicon optical modulators can be operated to function as (low efficiency) photovoltaic cells, where part of the input optical power is converted into useful photocurrent (Fig. 2b, Supplementary Discussion 2). If biased appropriately, such photocurrent can generate energy that can be put back into the system (gray shaded area in Fig. 2b). While this has been recognized before[19-21] an efficient and inexpensive method for harvesting photovoltaic energy in optical modulators has, to the best of the authors' knowledge, not been proposed nor realized.

In this work, we present an optical modulator capable of harnessing the energy generated by its photogenerated charges, which is used to self-charge the device capacitance. By adding a nanoscale electrical switch, efficient switching of the modulator between the nearly open circuit (or charged, $V_{mod} \approx 0.7$ V) and nearly short circuit conditions (or discharged, $V_{mod} = 0$ V) can be achieved. Changes in the modulator photovoltage translate into a change in the carrier density seen by the optical mode and hence

in the optical transmission through the device, achieving optical modulation. A model for such a device is presented, demonstrating optical switching at GHz speeds with the external electrical source consuming attojoule/bit energy (Fig. 2a). We also show how electrical energy consumption in the 20–100 zeptojoule/bite range is possible at MHz speeds, which approaches the Landauer limit for switching $E = k\text{Tln}(2) = 3\,\text{zJ}$[22]. We experimentally verify our model using a device fabricated in a commercial CMOS process (GlobalFoundries 45RFSOI) and show optical switching with an external electrical energy consumption as low as 20 zJ/bit at a data rate of 1 Mbps and an associated energy dissipation of 40 fJ/bit due to the photovoltaic effect.

## Results

**Operating principle and device model.** Figure 2c shows a typical configuration for driving an optical modulator through a CMOS inverter. Every time the input voltage level $V_{\text{in}}$ changes, the modulator capacitance $C_{\text{mod}}$ is charged by the external source $V_{\text{DD}}$ or discharged to ground, dissipating an average energy $E = C_{\text{mod}}V_{\text{DD}}{}^2/4$[19].

In contrast, our proposed modulation scheme, which we call photovoltaic (PV) modulation, relies on the photocurrent $I_{\text{pc}}$ (generated through optical absorption as discussed above) to provide the necessary energy to charge the modulator as depicted in Fig. 2d. Turning a transistor on and off through the voltage applied at its gate allows us to switch the voltage at the modulator terminals between 0 and the open-circuit voltage of the modulator $V_{\text{oc}}$. Thus, the only electrical energy supplied by the driving source in our scheme is that associated with the charging and discharging of the gate capacitance of the transistor $E = C_{\text{g}}V_{\text{in}}{}^2/4$. As we will discuss in detail below, our scheme achieves electrical gain, resulting in a reduction of the necessary input voltage $V_{\text{in}}$ and thus reduced energy dissipation.

The circuit model for the PV modulator—a diode-embedded optical modulator coupled to a transistor—is shown in Fig. 3a, which also shows the static (DC) operating characteristics. Good agreement between simulation (solid lines) and measurement (crosses) is observed. For low input voltages (low $V_{\text{gs}}$) the transistor is off (in a low conduction state $I_{\text{SW}} \approx 0$) and the voltage at the modulator $V_{\text{mod}}$ is the open-circuit voltage of the PV cell $V_{\text{oc}}$. As the input voltage rises, the transistor turns on until for $V_{\text{gs}} > V_{\text{th}}$ all the photocurrent flows through it, resulting in $V_{\text{mod}} \approx 0$.

The nonlinear electrical characteristic of the transistor results in the transition from the on-state to the off-state occurring rapidly for a narrow range of input voltages. Therefore, a small change in input voltage results in a large change in the voltage at the modulator terminals. This increased sensitivity translates into an increase in modulation efficiency and is explained due to the signal gain associated with a conventional common source transistor amplifier (Supplementary Discussion 3). At low frequencies, such voltage gain is given by $A_{\text{v}} = \Delta V_{\text{mod}}/\Delta V_{\text{gs}} = -g_{\text{m}}(r_0 || r_{\text{d}})$. Here, $g_{\text{m}} = dI_{\text{sw}}/dV_{\text{gs}}$ is the transistor transconductance, $r_0 = dV_{\text{mod}}/dI_{\text{sw}}$ the output resistance of the transistor, and $r_{\text{d}} = dV_{\text{mod}}/dI_{\text{mod}}$ the dynamic resistance of the diode.

The bias point at which the maximum gain $A_{\text{v}}$ is achieved is highlighted in Fig. 3a by the diamonds (simulated data) and circles (experimental measurements). As detailed in Supplementary Discussion 3, the maximum gain in our PV modulator corresponds to the so-called intrinsic gain of the transistor $|A_{\text{v,max}}| \approx g_{\text{m}}r_0$, which is $\approx 15$ in our 45 nm technology node as shown in the top plot of Fig. 3b. Note that, unlike conventional electrical amplifiers, this gain is achieved without the need of external electrical power, due to the self-biasing of the transistor through the photogenerated current $I_{\text{pc}}$. It is also important to recognize that there is essentially no energy

dissipation associated with the static bias voltage on the gate of the CMOS transistor since the leakage current is <pA.

Thus, our PV modulator achieves great reduction in electrical energy consumption because the voltage gain allows us to employ driving voltages 10–15× smaller than conventional modulators. We can write:

$$\frac{E_{\text{el,conventional}}}{E_{\text{el,PV}}} = \frac{\frac{1}{4}C_{\text{mod}}V_{\text{DD,conv}}^2 + I_{\text{pc}}V_{\text{bias}}}{\frac{1}{4}C_{\text{g}}V_{\text{pp,PV}}^2}$$
$$= \frac{\frac{1}{4}C_{\text{mod}}V_{\text{DD,conv}}^2 + I_{\text{pc}}V_{\text{bias}}}{\frac{1}{4}C_{\text{g}}\left(\frac{V_{\text{DD,conv}}}{A_{\text{v}}}\right)^2} \quad (1)$$

It is apparent from Eq. (1) that the device input capacitance ($C_{\text{mod}}$ in a conventional configuration and $C_{\text{g}}$ in a PV configuration) has a significant effect on electrical energy consumption. Typical values for the modulator capacitance are on the order of $C_{\text{mod}} > 15\,\text{fF}$. In contrast, the gate capacitance in modern CMOS processes is $C_{\text{gb}} \approx C_{\text{gs}} \approx C_{\text{gd}} \approx 0.1\text{–}1\,\text{fF}$ (Supplementary Discussion 4). Nevertheless, the electrical gain achieved in our PV modulator configuration increases the equivalent input capacitance through the Miller effect[23], resulting in $C_{\text{g}} \approx C_{\text{gs}} + C_{\text{gb}} + (1 + |A_{\text{v}}|)C_{\text{gd}} \approx 1\text{–}10\,\text{fF}$ (Supplementary Discussion 4). Notice how here we neglect the capacitance associated with the electrical pads or wiring, since these will not be significant for most systems which will utilize an on-chip driver.

From Eq. (1), disregarding the term $I_{\text{pc}}V_{\text{bias}}$ in conventional modulation and using $C_{\text{mod}} = 2C_{\text{g}}$ and $|A_{\text{v}}| = 10$, energy gains on the order of 200 are achievable. Using $V_{\text{DD,conv}} \approx 500\,\text{mV}$, driving voltages $V_{\text{pp,PV}} \approx 50\,\text{mV}$ are possible. Combined with a gate capacitance $C_{\text{g}} \approx 10\,\text{fF}$, electrical energy dissipations in the order of $E_{\text{el,PV}} = C_{\text{g}}V_{\text{pp,PV}}^2/4 \approx 5\,\text{aJ/bit}$ and lower are possible for the driving source.

There exists, of course, an additional source of electrical energy in our device, which is that associated with the charges generated from optical absorption, contributing an additional power dissipation $P_{\text{el,abs}} \approx I_{\text{pc}}V_{\text{mod}}$. As we will discuss below, such energy dissipation is on the order of 10–100 fJ/bit, and therefore dominates over the electrical energy supplied by the driving electrical source. This energy is, nevertheless, essentially free, as it is a result of the input optical power which is required for the operation of any optical modulator. While in conventional modulation such energy is wasted, here we use it to self-charge the device.

The PV modulator's speed is limited by the RC time constant at the modulator terminals, which is $\tau = (C_{\text{mod}} + C_{\text{ds}} + C_{\text{db}})(r_0 || r_{\text{d}}) \approx C_{\text{mod}}(r_0 || r_{\text{d}})$. Since the transistor output capacitance $C_{\text{ds}} + C_{\text{db}}$ is on the order of 0.5 fF or lower, we can approximate $C_{\text{mod}} + C_{\text{ds}} + C_{\text{db}} \approx C_{\text{mod}}$. The bandwidth at the maximum gain point is shown in the bottom plot of Fig. 3b. Greater bandwidths are obtained for larger photocurrents because a larger current $I_{\text{sw}}$ flows through the transistor, which results in lower $r_0$ since $r_0 \propto 1/I_{\text{sw}}$. For CMOS transistors in deep-submicron technology, typical values for $r_0$ are on the order of 0.1–10 MΩ (Supplementary Discussion 3), which results in bandwidths between 1 and 100 MHz.

A gain-bandwidth product for the PV modulator describes the inherent trade-off between improved modulation efficiency (sensitivity) and increased speed. The gain-bandwidth product can be expressed as:

$$|A_{\text{v}}|f_{\text{3dB}} = g_{\text{m}}(r_0 || r_{\text{d}})\frac{1}{2\pi(r_0 || r_{\text{d}})C_{\text{mod}}} = \frac{g_{\text{m}}}{2\pi C_{\text{mod}}} \quad (2)$$

We can thus bias the modulator at a point with a smaller electrical gain (and therefore increased energy dissipation), but larger bandwidth. Such trade-off is illustrated in Fig. 3c, which

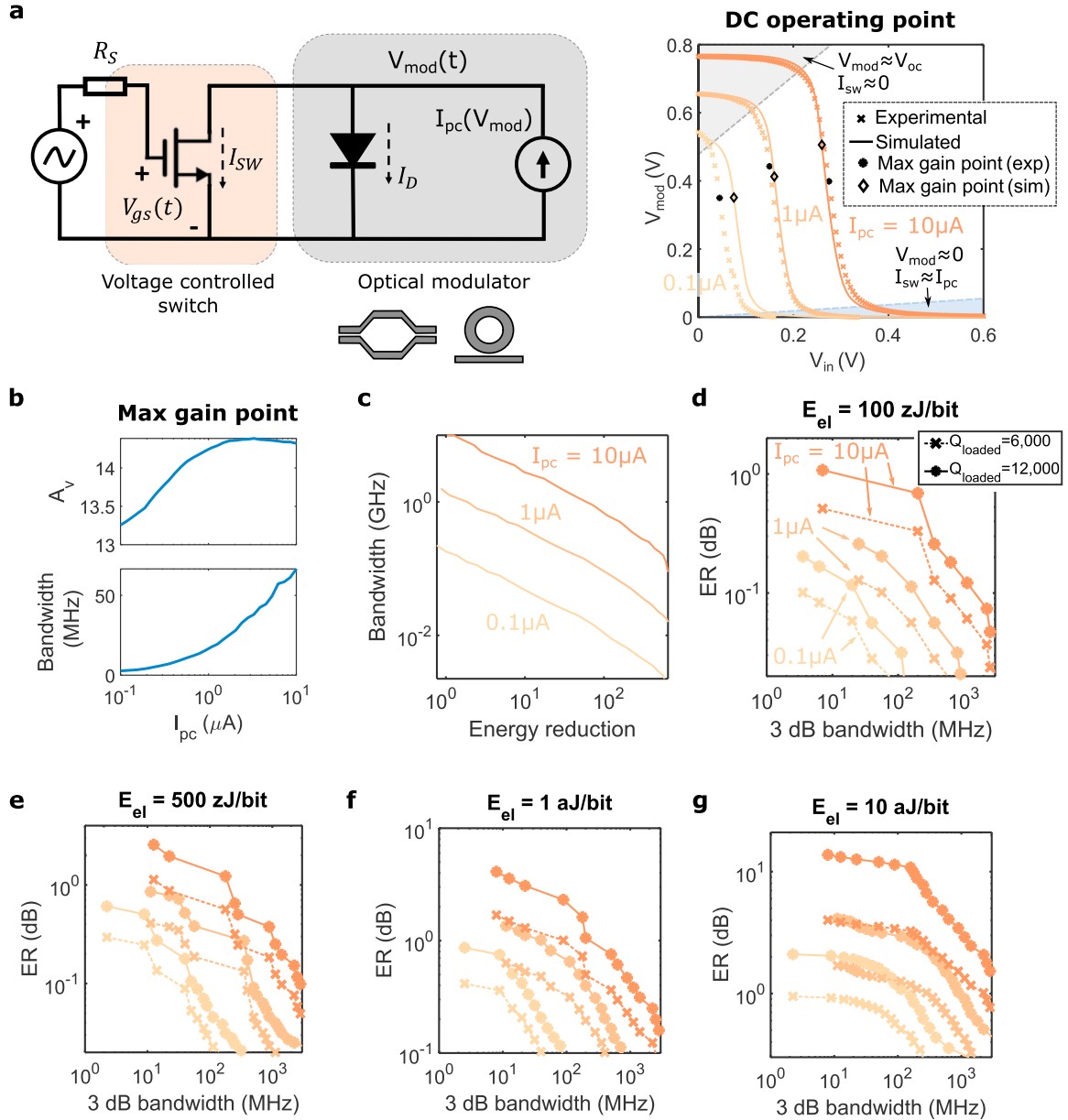

**Fig. 3 Photovoltaic modulator model and simulated performance. a** Large signal model of a photovoltaic (PV) modulator. By controlling the current flowing through the transistor ($I_{sw}$) via the gate voltage ($V_{gs}$) we can switch the voltage at the modulator terminals ($V_{mod}$) between 0 and the open-circuit voltage of the device $V_{oc}$. Simulation (lines) and experimental (crosses) data is shown. Diamonds and circles highlight the bias point resulting in maximum gain. **b** Voltage gain $A_v$ (top) and bandwidth (bottom) at the maximum gain point as a function of photogenerated current ($I_{pc}$). **c** Achievable bandwidth as a function of the reduction in electrical energy consumption compared to conventional modulation, calculated using Eq. (1). $C_{mod} = 20$ fF and $C_g$ is calculated as outlined in Supplementary Discussion 4. **d–g** Extinction ratio (ER) as a function of 3 dB bandwidth in a PV modulator for four different electrical energy consumptions ($E_{el}$). Crosses correspond to a Lorentzian resonance with a loaded quality factor $Q_{loaded} = 6000$, and dots to $Q_{loaded} = 12,000$. The maximum insertion loss of the modulation is limited to 6 dB in all plots. Different colors correspond to different generated photocurrents as indicated in (**d**).

shows the bandwidth as a function of the energy gain with respect to conventional modulation for different generated photocurrents $I_{pc}$. The energy gain is calculated using Eq. (1), assuming that the term $I_{pc}V_{bias}$ is a small contribution to the energy consumption, using $C_{mod} = 20$ fF and the value of $C_g$ calculated as outlined in Supplementary Discussion 4. Large energy gains above 100× are achievable with limited bandwidths in the order of 1–50 MHz. On the other hand, lower energy gains in the order of 10× can result in bandwidths above 1 GHz, which is desirable in applications requiring higher switching rates. Supplementary Discussion 3

shows additional data regarding the frequency response of PV modulators.

As described in Supplementary Discussion 5, from the calculated voltage swing at the modulator terminals $\Delta V_{mod}$ we can extract the change in the optical transmission through the modulator and from that the optical modulation characteristics—namely extinction ratio (ER) and insertion loss (IL). Figure 3d–g show the simulated ER for four different electrical energy dissipations, assuming a close to critically coupled Lorentzian resonance with loaded Q factors $Q_{loaded} = 6000$ and $Q_{loaded} = $

12,000 and an intrinsic modulation efficiency based on representative experimental results. In all cases, the IL is limited to be lower than 6 dB. As expected, larger $Q$ factors result in better ER, since a stronger transmission change is obtained for the same resonance shift $\Delta\lambda_0$. Operation at higher speeds results in decreased ER due to the gain-bandwidth trade-off discussed above.

ER in the order of 1 dB (20% change in transmission) can be achieved at bandwidths close to 10 MHz for low electrical energy consumptions between 100 and 500 zJ/bit (Fig. 3d, e). While relatively low, such ER is enough to achieve close to error-free communication at these data rates (Supplementary Discussion 6). For comparison, the same microring device with the same modulation efficiency would only achieve ER ≈ 0.04 dB if modulated with a 100 zJ/bit electrical energy using the conventional carrier depletion (reverse-biased) approach. For the PV modulator at energies >1 aJ/bit (Fig. 3f, g), ER > 2 dB at speeds approaching 1 GHz can be achieved.

**Experimental demonstration.** We designed and implemented monolithic PV modulators in an unmodified 45 nm commercial microelectronics CMOS process (GlobalFoundries 45RFSOI, see "Methods"). A micrograph and the layout of the device are shown in Fig. 4a, b. We used a microring modulator with 5 μm outer diameter and 1.2 μm width, designed for operation at wavelengths around 1270 nm[24]. The PV modulator uses interdigitated PN junctions around the circumference of the microring. While the intrinsic absorption of silicon could be used for PV modulation, a ring of silicon germanium (SiGe) with bandgap wavelength of $\lambda_{bandgap} \approx 1215$ nm was embedded to enhance optical absorption and the photovoltaic effect[25]. The SiGe ring is 300 nm wide and has partial overlap with the microring optical mode, resulting in a measured responsivity of $R = 34$ mA/W.

We used a body-contacted NMOS transistor for modulating the photogenerated charge in the microring device, with nominal gate length and width of 56 nm and 2.4 μm, respectively. From the electrical model provided by the foundry, the input gate capacitances are $C_{gb} \approx C_{gs} \approx C_{gd} \approx 0.5$ fF (Supplementary Discussion 4). Minimum width transistors in the same process have smaller input capacitances on the order of 0.1 fF, which would result in decreased energy consumption. Notice how the low voltage levels present in our PV modulator allow for the use of core transistors with the minimum available oxide thickness and gate length, which would not be possible in conventional modulation.

Transmission spectra of the device for different bias voltages applied to the transistor gate are shown in Fig. 4c. As the transistor increasingly turns on, the voltage at the modulator terminals $V_{mod}$ decreases (Fig. 3a), resulting in a reduction in the free-carrier density and a blue shift in the resonance wavelength. Supplementary Discussion 7 shows similar data for a Si-only device without embedded SiGe.

The measured frequency response of the PV modulator at the maximum gain point for different generated photocurrents is shown in Fig. 4d. As discussed, we observe an increase in the 3 dB bandwidth when photocurrent is increased: from 4.6 MHz for $I_{pc}$ = 0.22 μA to 35 MHz for $I_{pc}$ = 3.2 μA. There is good agreement between experimental and simulation results. The dashed black line shows the frequency response of the same device operated in a conventional, reverse-biased configuration with the same electrical energy dissipation. As expected, the response at low frequencies is significantly weaker than that of the PV modulator due to the absence of voltage gain, but the frequency response is faster with a 3 dB bandwidth close to 5 GHz.

When biased at the point of maximum modulation efficiency, a 100 mV$_{PP}$ peak-to-peak voltage signal applied to the transistor gate results in 0.6 dB ER with 7.5 μW on-chip input optical power, requiring only 8.75 aJ/bit electrical energy to be delivered by the electrical driving source (Fig. 4e). Observable modulation can be achieved with input peak to peak voltages as low as 4 mV$_{PP}$, corresponding to a 23.2 zJ/bit electrical energy consumption (Fig. 4f). The IL is lower than 3 dB, which is comparable to the IL of most modern optical modulators (Supplementary Tables 1 and 2). Figure 4g shows a modulation waveform obtained with conventional reverse bias modulation. A smaller ER of about 0.05 dB is obtained for a 100 mV$_{PP}$ driving signal, corresponding to an energy consumption of 50 aJ/bit (5.5× higher than when using PV modulation due to the larger input capacitance). As described in Supplementary Discussion 6, the experimentally achieved SNR for the waveforms in Fig. 4e–g is limited by the high optical loss (~10 dB per coupler) of the grating couplers providing optical access to the device.

## Discussion

In this work, we have experimentally demonstrated optical modulation with externally supplied electrical energies as low as 25 zJ/bit at 1 MHz data rate. Such an electrical energy consumption approaches the Landauer limit for the minimum energy necessary for binary switching[22], which is given by $kT\ln(2) = 3$ zJ at 300 K. As already mentioned, the total electrical energy dissipation in our device is not only provided by the external electrical driver but includes also the energy generated by the photovoltaic effect. An on-chip optical power of 7.5 μW generates a 0.25 μA photocurrent, and the operating voltage is $V_{mod} \approx 0.15$ V at the maximum gain point. This corresponds to a photovoltaic electrical power of 40 nW, resulting in a 40 fJ/bit energy dissipation at 1 Mbps. This energy is not supplied by an external electrical source but recovered from the sub-bandgap optical absorption occurring in the device, which is unused and dissipated as heat in conventional modulation. Of course, our photovoltaic modulator has also an associated optical power dissipation. At 1 Mbps operation, the required 7.5 μW input optical power translates into a 7.5 pJ/bit optical energy consumption (notice that only 50% of this is consumed in the transmitter, since we operate at IL close to 3 dB).

It is interesting to analyze the performance of our modulation scheme in terms of the number of electrons that the external driver needs to provide to achieve optical switching. For a 200 mV$_{PP}$ (20 mV$_{PP}$) driving voltage, corresponding to a 25 aJ/bit (550 zJ/bit) electrical energy dissipation, only 2500 (550) electrons need to be provided by the external driver. In contrast, the lowest power Si modulator reported in the literature[26] requires 50,000 electrons, and the lowest power integrated optical modulator in other material platforms[14] requires 100,000.

Improved performance in terms of device speed and total power dissipation could be obtained by enhancing the responsivity of the device (with the measured responsivity of 0.034 A/W, only 9% of the internal loss contributes to photocurrent generation), and higher ER could be achieved by increasing its $Q$ factor (active silicon ring resonators with Q factors as large as 40,000 have been demonstrated[27]). Similarly, reductions in the output resistance of the transistor or decreased modulator capacitance would enable higher speed operation. Circuit techniques capable of storing and reusing the charge that is shunted to the ground when the transistor turns on could further increase the energy efficiency of our device.

In conclusion, in this work, we have presented and experimentally characterized an operational regime for optical modulators which we call the photovoltaic regime. We leverage the parasitic photocurrent generated by semiconductor optical modulators when light is input to the device to bias a transistor at

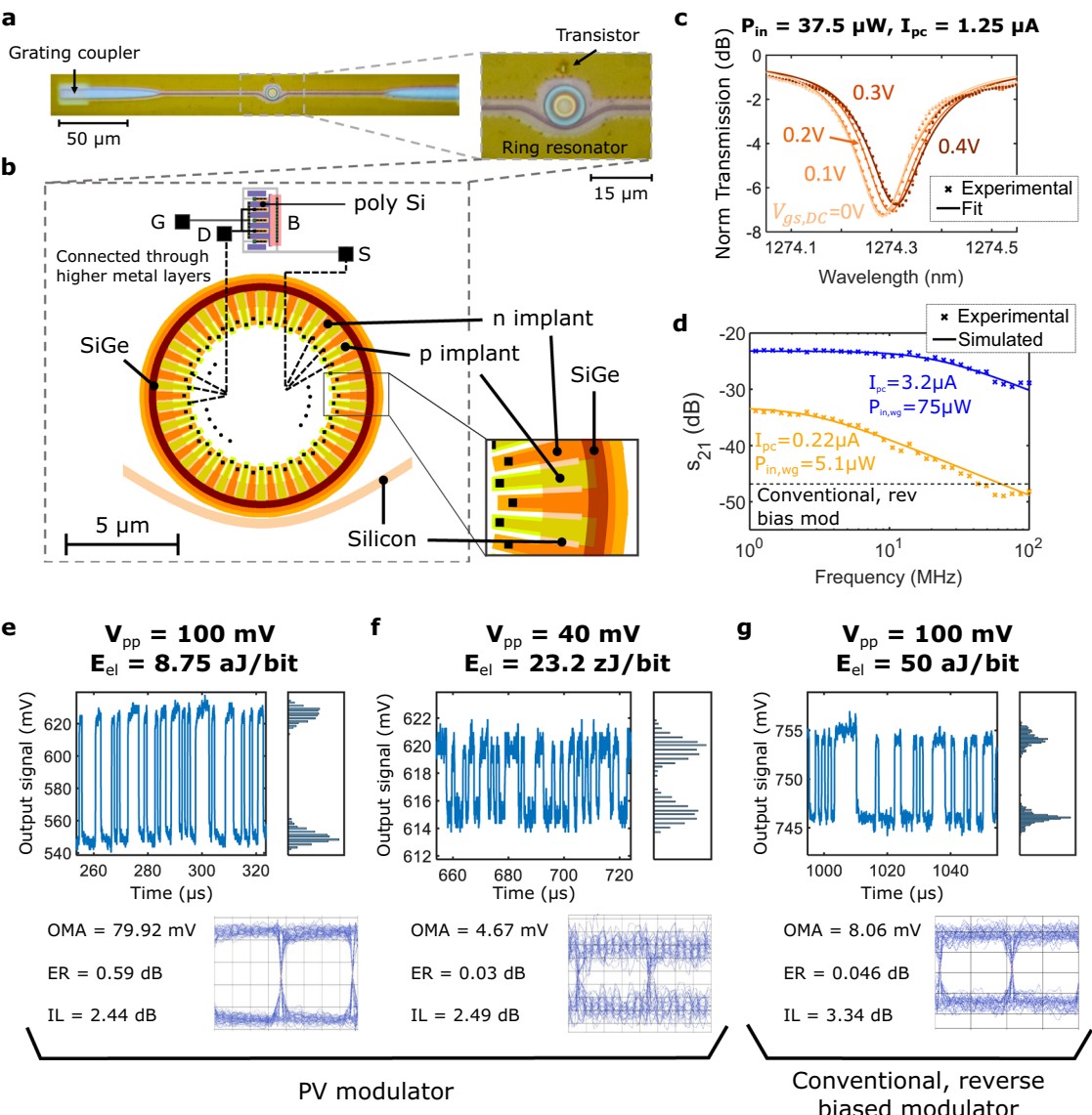

**Fig. 4 Experimental characterization of a photovoltaic (PV) modulator. a** Micrograph of the fabricated device. The ring resonator modulator, the transistor, and the grating couplers providing optical access to the chip are visible. **b** Device layout, with a closeup on the T junction design for the modulator spokes. Yellow (orange) shading corresponds to p-doped (n-doped) regions, and the silicon-germanium (SiGe) band is shown in brown (see "Methods"). **c** DC transmission spectra for different gate bias voltages ($V_{gs,DC}$) for 37.5 μW on-chip optical power, corresponding to a photogenerated current $I_{pc} = 1.25$ μA. **d** PV modulator frequency response for $I_{pc} = 0.22$ μA (yellow) and $I_{pc} = 3.2$ μA (blue). Crosses correspond to experimentally measured values, while solid lines are simulation results. The dashed black line shows the response of a conventional, reverse-biased modulator. **e, f** Output optical signals for the device operated in PV mode with 100 mV$_{pp}$ (**e**) and 4 mV$_{pp}$ (**f**) driving signals. The histogram of the detected '1' and '0' bits, its corresponding eye diagram and the modulation characteristics are shown (OMA optical modulation amplitude, ER extinction ratio, IL insertion loss). The data rate is 1 Mbps and the on-chip optical power is 7.5 μW, corresponding to $I_{pc} = 0.25$ μA. **g** Output optical signal for the device operated in a conventional configuration with a 100 mV$_{pp}$ driving signal and $-2$ V reverse bias. The data rate is 1 Mbps.

a point where voltage amplification is achieved, therefore reducing the necessary driving electrical signal and the electrical energy necessary to drive the device. The optimal bias point for the transistor used for PV modulation is in the weak-inversion or sub-threshold regime. Similar to the energy-bandwidth trade-off that is well established for sub-threshold logic circuits, achieving the lowest possible electrical energy consumption for optical modulation comes at the cost of limited operation speed[28].

Besides the decrease in the required peak-to-peak driving voltage, further energy reduction is obtained because the input capacitance that the external source drives is now that of a nanoscale transistor. Even when accounting for the increase in input capacitance due to the Miller effect (Supplementary Discussion 4) the gate capacitance is between 500 aF and 5 fF, and 4–40× lower than the capacitance of the micrometer scale modulator. This small capacitance seen by the driving source is, in fact, comparable to that of a simple metal trace (~0.2 fF/μm). Photovoltaic optical modulation therefore approaches the practical limit to electrical energy consumption.

Bandwidths approaching 1 GHz with 10 aJ/bit of electrical energy consumed by the driving source can be achieved, which is 100× lower than any silicon modulator reported to date. We also showed how optical modulation with moderate speeds on the order of 1–10 MHz and 0.5–1 dB ER is achievable with driving

electrical energies as low as 50 zJ/bit. The reduction in electrical energy consumption does not require additional optical power relative to conventional modulation—it simply puts to use the parasitic photocurrent which would otherwise be wasted. While electrical energy consumption may not dominate every transmitter (Supplementary Tables 1 and 2), our scheme lowers the total power consumption for any silicon modulator by using the photocurrent to reduce the electrical energy consumption. These dramatic improvements come at the expense of lower bandwidths than conventional modulation schemes, and hence are most relevant when the necessary signal bandwidth is sub-GHz.

As discussed in the introduction, the photovoltaic modulation scheme with low-power consumption could find applications where the electrical energy supply is constrained. The experimentally demonstrated optical modulator operates with a total power dissipation (optical and electrical) of <8 μW, and only requires 8.75 pW of electrical energy from the electrical driving source for 1 Mbps operation, making it possible for an electrical sensor to directly drive the modulator without the need for amplification (Fig. 1). Such data rate and power offer significant improvements over the state-of-the-art and may enable 1000's of high-density, parallel interconnects.

## Methods

**Device models.** The transistor model, which is based on the BSIM MOSFET model, was provided by the Process Design Kit (PDK) of the CMOS process we use to fabricate our devices (GlobalFoundries 45 nm RF SOI). For the modulator, we used the standard SPICE diode model[29] with representative values for our CMOS resonant modulators (Supplementary Table 3).

**Device fabrication.** The device was fabricated in GlobalFoundries 45RFSOI process, a commercial high-performance 45 nm complementary metal–oxide–semiconductor (CMOS) silicon-on-insulator (SOI) process. The ring is fabricated in the crystalline silicon layer, which is conventionally used to realize the transistor body, source, and drain. To generate the interleaved p–n junctions we use standard CMOS doping implants, which are normally used for the generation of the different transistor regions (examples of such doping implants include source/drain, halo, source/drain extension, body…).

Our design complies with all the foundry design rules, therefore allowing for the fabrication of our photonic device without any modification to the microelectronics process flow, in what is commonly referred to as the zero-change CMOS approach[30]. This results in a low-cost, highly scalable photonic platform that can be monolithically integrated with electronics[31], which is essential to realize our PV modulator scheme without incurring in large performance penalties due to parasitic effects. The SiGe implant used in our design to increase photocurrent generation is also a standard layer in such processes, which is used to strain the channel of p-FETs to improve transistor speed. Our design uses T-shaped spokes to realize the interleaved p–n junctions to decrease parasitic capacitance and increase device speed[32]. Our resonant modulator design is similar to that reported in ref. [24].

**Experimental setup.** To experimentally characterize the device, the light generated by a tunable O band laser (Agilent 81600B) was coupled into and out of the chip through optical fibers (SMF28), which were aligned to the vertical grating couplers in the chip via 3-axis stages.

To record transmission spectra (Fig. 4c), the optical power coupled to the output optical fiber was monitored with a power meter (Agilent 81635A) as the laser wavelength was swept. A source meter (Keithley 2400) was used to apply the necessary bias voltage to the gate of the transistor, which was contacted via a 50 μm pitch ground-signal (GS) electrical probe (Cascade Microtech GS-50).

To measure the device bandwidth (Fig. 4d), a single frequency sinusoidal signal generated by an arbitrary waveform generator (AWG, Agilent 81180A) was applied to the transistor gate. The optical signal generated by the modulator was then coupled to a photodetector (Thorlabs DX20AF) and subsequently connected to a microwave spectrum analyzer (MSA, HP70900A). By sweeping the frequency $f_0$ of the applied sinusoid and measuring the electrical power of the photodetected signal at that same frequency $f_0$ (with the MSA) we obtained the frequency response of the device.

To record modulation waveforms (Fig. 4e–g), the AWG was used to supply a pseudo-random binary sequence of length $L = 2^7 − 1$ (PRBS7) to the transistor gate. The optical signal generated by the modulator was amplified using a semiconductor optical amplifier (SOA, Thorlabs BOA1130S), and its output connected to a clean-up tunable optical filter to remove out of band amplified spontaneous emission (ASE) noise. Optical amplification was necessary to overcome the high optical loss (~10 dB) associated with outcoupling of the light

through the vertical grating coupler, allowing to generate a large enough signal to be observed in an oscilloscope (Infiniium 54833A DSO).

The described setups are depicted in Supplementary Fig. 9.

## Data availability

The data that support the findings of this study are available from the corresponding author upon request.

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

## Acknowledgements

We thank Darío de la Fuente for his help with initial measurements, and Bohan Zhang and Hayk Gevorgyan for providing grating coupler designs. We acknowledge support from the Office of the Director of National Intelligence (ODNI), Intelligence Advanced Research Projects Activity (IARPA), via the U.S. Army Research Office (ARO) grant W911NF-19-2-0114. M.C. is partially funded by La Caixa Foundation under award LCF-BQ-AA17-11610001.

## Author contributions

M.C. designed the device, derived the equivalent circuit model, performed the experimental characterization, and wrote the manuscript. A.A. proposed the device concept and performed initial tests. R.R. supervised the project.

## Competing interests

R.R. is developing silicon photonic technologies at Ayar Labs, Inc. The remaining authors declare no competing interests.
