## [Peer Review File · Nature Communications]

REVIEWER COMMENTS

Reviewer #1 (Remarks to the Author):

The authors demonstrate a "photovoltaic" ring modulator operating with record electrical energy efficiency by operating a silicon P-N diode in weak forward bias. The novelty of their work lies in the "recycling" of the photocurrent in the silicon diode to self-bias or charge a transistor, which in turn can switch the driving voltage of the diode with some electrical gain. The trade-off is a reduction in electrical bandwidth due to large output resistance of the transistor, which limits the device to operation in the MHz regime. A few comments and questions from me below:

- What is the waveguide loss for the SiGe waveguide used in the modulator?
- It may be helpful to point out that I_{pc} in Fig 2(c) and (d) has an opposite sign compared with I in Fig 2(b).
- In Fig. 3, figures (d) through (g) are not very clear in terms of which line corresponds to which photocurrent. Also, better visual clarity between the dots and x's would be appreciated.
- In comparing rings with loaded $Q = 6000$ and 12000 , the authors comment that an increase in extinction ratio is caused by a steeper slope in the ring transfer function. The change in Q should also affect the responsivity of the ring, which would in turn change the photocurrent. Is this also considered in the simulations?
- What V_{gs} bias is used for the experimental results in Fig. 4? Was the bias chosen such that the electric gain is the maximum, or for optimal modulation efficiency?
- It seems that insertion loss in this paper is defined as the "1" level of the modulator. Is this consistent with all the results tabulated in the supplementary document? Sometimes, people will quote the insertion loss as the average power.

Overall, the paper is very well written and the supplementary information contains very thorough background review and supporting data. The underlying concept is very interesting, although applications may be ultimately limited by the relatively slower speeds, and the fact that the energy associated with the modulator may be dwarfed by other things such as laser power. Nonetheless, I believe this paper to be an excellent contribution to Nature Communications.

Reviewer #2 (Remarks to the Author):

The manuscript by de Cea et al. describes Si modulators that use a photovoltaic (PV) effect to reduce the energy consumption of modulation. The proposal is interesting, but the manuscript should be revised before it can be acceptable for publication. Below are my comments.

1. For the material to exhibit a PV effect at the wavelength of operation, it must be optically absorbing to some extent. Absorption and voltage-dependent absorption are parasitic for a phase-modulator and contributes to additional chirp. Would the authors please discuss more about this trade-off between loss and PV in the main manuscript? Should a device designer try to eliminate parasitic losses, in which case this proposed effect would not exist? How much absorption can be tolerated?

2. In Fig. 1, among the proposed applications, some are not reasonable for Si waveguides. Please revise or clarify these use cases. For brain (optogenetic) applications (b), the operation wavelength is often blue or green, so Si waveguides and modulators are too absorptive and are not used. For single photon applications (c), is having absorption in the modulator better than striving to reduce loss (see Q.1)? The MRI application in (d) might be fine, but it may require higher extinction ratios than demonstrated in this article.

3. On page 3, it is stated that parasitic absorption and free carrier generation is universal. Does

this also occur in insulator materials like LiNbO₃ or BaTiO₃? To my knowledge, there is almost no parasitic absorption. The bandgap is also very large. The authors' claim may only be most applicable to semiconductors. Would the authors please clarify to substantiate their claim (please include references), or revise?

4. Surface passivation would reduce the photocurrent. It is not clear how general the results in this article can be adopted by others in the field who use different processes to fabricate the devices.

5. Despite the authors stating that the parasitic absorption of Si in the infrared could be used for the PV modulation, the device demonstrated has Ge to increase the absorption near 1300nm wavelength! The introduction and motivation for this article should be revised to not over-extend the proposal, claims, and the results.

6. Would the authors please clarify the device insertion loss quoted in Fig. 4? The quoted IL is 2.4dB-3.34dB. Does this also include material loss?

7. The achieved ER is < 0.6 dB, which is very low and is not usually acceptable for optical communications. Page 9 states a 1dB ER is sufficient for error free communication. What is reference for this claim? In optical communication, the bit error rate (BER) is the important metric. A minimum BER of 10^{-3} is needed for forward error correction (FEC). What is the achieved BER here? The use of FEC will significantly increase overall power consumption, thus eliminating any benefits of this PV modulation scheme.

8. There is indeed a great deal of interest in minimizing the energy consumption of optical transmitters to fJ/bit levels or less. But this energy consumption specification includes the laser source and the driver electronics, which are often the dominant source of energy consumption and not the modulator itself. As the power dissipation of a laser is static, modulating at high bit rates has the advantage of reducing the energy per bit (= power / bit rate). In this article, the power consumption of the laser has been neglected. It would be instructive to quote the total energy per bit (including the laser source).

9. While this article has an interesting proposal, it seems to have missed the jugular when one focuses on energy consumption. I would recommend reframing the motivation of the article and presenting the work in a more balanced way that describes both the advantages and limitations of the work.

Reviewer #1

The authors demonstrate a “photovoltaic” ring modulator operating with record electrical energy efficiency by operating a silicon P-N diode in weak forward bias. The novelty of their work lies in the “recycling” of the photocurrent in the silicon diode to self-bias or charge a transistor, which in turn can switch the driving voltage of the diode with some electrical gain. The trade-off is a reduction in electrical bandwidth due to large output resistance of the transistor, which limits the device to operation in the MHz regime. A few comments and questions from me below:

- What is the waveguide loss for the SiGe waveguide used in the modulator?

Response: The total waveguide loss can be extracted from the quality factor and extinction ratio of the ring resonator. For our devices, the ring waveguide loss is about 6.34 cm^{-1} (27.5 dB/cm).

From preliminary measurements characterizing the device, 0.6 cm^{-1} (2.6 dB/cm) of the total loss correspond to photocurrent-generating absorption. The remaining losses ($5.74 \text{ cm}^{-1} = 24.9 \text{ dB/cm}$) can be mainly attributed to free carrier absorption.

This information has been added in Supplementary Section S2.

- It may be helpful to point out that I_{pc} in Fig 2(c) and (d) has an opposite sign compared with I in Fig 2(b).

Response: An inset in Fig 2(b) showing the I and V definitions has been added to clarify the current signs.

- In Fig. 3, figures (d) through (g) are not very clear in terms of which line corresponds to which photocurrent. Also, better visual clarity between the dots and x's would be appreciated.

Response: Figs. 3(d) through (g) have been modified for easier visualization.

- In comparing rings with loaded $Q = 6000$ and 12000 , the authors comment that an increase in extinction ratio is caused by a steeper slope in the ring transfer function. The change in Q should also affect the responsivity of the ring, which would in turn change the photocurrent. Is this also considered in the simulations?

Response: It is true that, in general, the Q factor of the ring will affect the responsivity of the device. However, our results are given as a function of generated photocurrent. Therefore, the conversion from input optical power to generated photocurrent (through the responsivity R) is already incorporated here.

- What V_{gs} bias is used for the experimental results in Fig. 4? Was the bias chosen such that the electric gain is the maximum, or for optimal modulation efficiency?

Response: For operation of the modulator, we ultimately care about maximizing the amplitude of the generated modulation signal. Thus, V_{gs} was chosen to maximize modulation efficiency. A sentence has been added pointing to this fact (line 237 in the main manuscript).

- It seems that insertion loss in this paper is defined as the “1” level of the modulator. Is this consistent with all the results tabulated in the supplementary document? Sometimes, people will quote the insertion loss as the average power.

Response: To the best of our knowledge, all quoted insertion losses in tables 1 and 2 are defined as the “1” level of the modulator.

Overall, the paper is very well written and the supplementary information contains very thorough background review and supporting data. The underlying concept is very interesting, although applications may be ultimately limited by the relatively slower speeds, and the fact that the energy associated with the modulator may be dwarfed by other things such as laser power. Nonetheless, I believe this paper to be an excellent contribution to Nature Communications.

Reviewer #2:

The manuscript by de Cea et al. describes Si modulators that use a photovoltaic (PV) effect to reduce the energy consumption of modulation. The proposal is interesting, but the manuscript should be revised before the it can be acceptable for publication. Below are my comments.

1. For the material to exhibit a PV effect at the wavelength of operation, it must be optically absorbing to some extent. Absorption and voltage-dependent absorption are parasitic for a phase-modulator and contributes to additional chirp. Would the authors please discuss more about this trade-off between loss and PV in the main manuscript? Should a device designer try to eliminate parasitic losses, in which case this proposed effect would not exist? How much absorption can be tolerated?

Response: The reviewer is absolutely correct in pointing out that some extent of optical loss is required for our scheme to work.

As discussed in Supplementary Section S2, it is not possible to entirely eliminate these parasitic effects in semiconductor devices. This has been recognized by several researchers and reported in a variety of publications (indicated in Supplementary Section S2), which observe these effects in fabrication processes specifically tailored for photonics. The main point of our work is recognizing the presence of this photocurrent and, instead of just treating it as a parasitic (i.e, undesired) effect, use it to our advantage. It is also worth pointing out that there have been other works that take advantage of parasitic optical absorption in semiconductor optical waveguides. For instance, the photocurrent has been used to monitor the optical power in a waveguide [Morichetti et al., *IEEE J. Sel. Top. Quantum Electron* 20, 292 (2014)] and to stabilize a microring resonator [Jayatilika et al., *Optics Exp.* 23, 25084 (2015)].

While for a conventional modulator configuration it is desirable to eliminate absorption, our work demonstrates how, in specific situations, having optical absorption (and even adding it intentionally) can enable devices and systems with better performance than state-of-the-art solutions. Examples include the applications proposed in the manuscript (Fig. 1). We believe that, if a low data rate link with low power consumption is desired, using our photovoltaic modulator is a promising approach and could be advantageous when compared to the use of electrical links or a scheme with a conventional modulator – where the parasitic photocurrent could be smaller but wasted and not used to effectively bootstrap an electrical amplifier as it is here.

2. In Fig. 1, among the proposed applications, some are not reasonable for Si waveguides. Please revise or clarify these use cases. For brain (optogenetic) applications (b), the operation wavelength is often blue or green, so Si waveguides and modulators are too absorptive and are not used. For single photon applications (c), is having absorption in the modulator better than striving to reduce loss (see Q.1)? The MRI application in (d) might be fine, but it may require higher extinction ratios than demonstrated in this article.

Response: In all of the examples provided, the photovoltaic modulator is used to replace the electrical readout link between the sensor and a remote computational processor. We will address each of the applications separately:

1. Electrophysiological sensing and neural interfaces: The neural interfaces we are exploring include a wide range of devices used to monitor neural/electrophysiological signals, and the PV modulator is used to establish an optical link between the sensing site and the processor. This does not include optogenetic techniques, which are used to stimulate genetically modified neurons through light.

Neural interfaces with both wired and wireless links have been reported in the literature to establish communication between the sensor and the processor. To date, such demonstrations have a relatively high power dissipation that limits the number of parallel sensing sites that can be supported without affecting the brain or any other tissue under study:

- Wired solutions have on the order of 200 pJ/bit or higher power dissipation due to the need of amplifying and digitizing the sensed signals. For instance, data available on the Neuralink probes [Musk et al., *J. Med. Internet Res.* 21, e16194 (2019)] reports a 5.2 μ W analog power dissipation, which at the maximum operating bandwidth of 27 kHz translates into 190 pJ/bit energy dissipation. The work in [Obaid et al., *Sci. Adv.* 6, eaay2789 (2020)] has 70 mW total power dissipation with a 20 kHz bandwidth, which translates into 280 μ W/channel and a 14 nJ/bit energy dissipation.
- Wireless solutions have demonstrated energy dissipations limited to about 20 pJ/bit and above due to the need of overcoming the propagation loss of the signal through the tissue. Moreover, wireless solutions have limited reach due to propagation and diffraction losses. The reviewer is pointed to [de Marcellis et al., *IEEE Trans. Biomed. Circuits Syst.* 14, 441 (2020)] for a compendium of reported wireless link demonstrations for neural interfaces.

Our demonstrated solution with a total (electrical + optical) power of 7.5 μ W (and an energy consumption of 7.5 pJ/bit at 1 MHz) could enable the readout of thousands of parallel sensors while still complying with the power dissipation limits set by the brain tissue.

2. Single photon applications: the purpose of the photovoltaic modulator in this application is not to transduce or transport the single photons (this is done by the superconducting nanowire single photon detector or SNSPD), but to substitute the electrical link that is usually employed to read out the signal generated by the SNSPD by an optical link. In this scenario, the PV modulator converts the electrical signal generated by the SNSPD to the optical domain. The reviewer is kindly pointed to [de Cea et al., *Sci. Rep.* 10, 9470 (2020)] for a more detailed discussion of the benefits of the use of optical readout in cryogenic applications.

As discussed in [de Cea et al., *Sci. Rep.* 10, 9470 (2020)], a cryogenic optical readout scheme will be beneficial as long as its power dissipation is lower than the electrical link alternative. The PV modulator could represent a significant improvement compared to previous demonstrations of optical readout because it eliminates DC electrical power dissipation. As long as this reduction in

power consumption is larger than the added absorption loss, it will be beneficial to use our PV modulator approach.

We have modified the first three paragraphs of the introduction to clarify our envisioned applications and clarify that the purpose of the PV modulator is to enable an optical link between the sensing site and the main processor.

3. On page 3, it is stated that parasitic absorption and free carrier generation is universal. Does this also occur in insulator materials like LiNbO₃ or BaTiO₃? To my knowledge, there is almost no parasitic absorption. The bandgap is also very large. The authors' claim may only be most applicable to semiconductors. Would the authors please clarify to substantiate their claim (please include references), or revise?

Response: The sentence in page 3 (line 74) has been modified to make it clear that we are referring to semiconductor-based optical modulators only. As the reviewer correctly points out, free carrier generation in insulator materials should be negligible.

4. Surface passivation would reduce the photocurrent. It is not clear how general the results in this article can be adopted by others in the field who use different processes to fabricate the devices.

Response: It is true that the quality of surface passivation will affect the strength of surface state absorption (SSA) and the generated photocurrent. Nevertheless, we believe that SSA is present in any modern fabrication processes, even for those with the highest passivation quality. Two facts support this claim:

1. As discussed in detail in Supplementary Section S2, several researchers using different fabrication processes and different waveguide geometries have reported the observation of SSA. These processes constitute a representative sample of photonics fabrication: CMOS foundries (our work), photonics-only foundries [Gil-Molina et al., *Appl. Phys. Lett* 112 (2018)] and in-house fabrication [Baehr-Jones et al., *Opt Express* 16 (2008)].
2. The silicon surface passivation in our CMOS process (45RF SOI from GlobalFoundries) is highly optimized. Since ours is a microelectronics process that is designed to yield billions of high-performance transistors, high quality surface passivation is needed to ensure the best, most consistent transistor performance. Low quality passivation results in transistors with decreased mobility and transconductance, which is unacceptable [Hamaide et al., *J. Appl. Phys.* 101, 114513 (2007)]. The crystalline Si layer in our fabrication process is passivated by SiN, which has been shown to result in excellent surface properties [Chowdhury et al., *Appl. Phys. Lett.* 101, 021601 (2012)].

As we report in the Supplemental Material the measured photocurrent in our all-silicon modulators is consistent with the expected photoresponse in the most advanced photonics processes.

5. Despite the authors stating that the parasitic absorption of Si in the infrared could be used for the PV modulation, the device demonstrated has Ge to increase the absorption near 1300nm wavelength! The introduction and motivation for this article should be revised to not over-extend the proposal, claims, and the results.

Response: The experimental results shown in the main part of the manuscript correspond to a device with embedded SiGe (with a Germanium concentration between 20% and 30 %, not pure Ge). Nevertheless, our approach is extendable to any semiconductor-based modulator that shows parasitic free carrier absorption, which includes Si and InP. **We show experimental data demonstrating the photovoltaic modulation approach working for a Si-only modulator in Supplementary Section S7** (Supplementary Section S6 in the original manuscript), showing a responsivity of 0.85 mA/W at a 1550 nm wavelength. While this responsivity is lower than the 34 mA/W measured in the SiGe modulator, the generated photocurrents are enough to obtain a useful shift in the resonance wavelength of the device (as shown in Supplementary Figure S7).

The analysis and simulations carried out in the “Operating principle and device modeling” section of the main manuscript are general and apply to any modulator that has a parasitic photocurrent in response of an input optical power, and are not particular to the Si resonator with or without embedded SiGe.

6. Would the authors please clarify the device insertion loss quoted in Fig. 4? The quoted IL is 2.4dB-3.34dB. Does this also include material loss?

Response: We use the standard definition of insertion loss in communications applications, which defines the insertion loss as the ratio between the output power of the ‘1’ bit and the output power of the device when it is out of resonance. This definition does not account for the losses of coupling in and out of the chip, but accounts for any material loss in the ring resonator.

The total waveguide loss can be extracted from the quality factor and extinction ratio of the ring resonator. For our devices, the ring waveguide loss is about 6.34 cm^{-1} (27.5 dB/cm).

From preliminary measurements characterizing the device, 0.6 cm^{-1} (2.6 dB/cm) of the total loss correspond to photocurrent-generating absorption. The remaining losses ($5.74 \text{ cm}^{-1} = 24.9 \text{ dB/cm}$) can be attributed to free carrier absorption.

This information has been added in Supplementary Section S2.

7. The achieved ER is < 0.6 dB, which is very low and is not usually acceptable for optical communications. Page 9 states a 1dB ER is sufficient for error free communication. What is reference for this claim? In optical communication, the bit error rate (BER) is the important metric. A minimum BER of 10^{-3} is needed for forward error correction (FEC). What is the achieved BER here? The use of FEC will significantly increase overall power consumption, thus eliminating any benefits of this PV modulation scheme.

Response: In typical high bandwidth optical communications applications the reviewer’s comment is correct: 0.6 dB ER is usually not acceptable (in fact, most standards require an ER > 2 dB).

Nevertheless, operation at the bandwidths that our target applications require significantly change the performance tradeoffs. The low receiver bandwidth substantially decreases the equivalent noise power of the receiver chain detecting the modulated signal generated by our PV modulator. This allows for the achievement of close to error-free transmission without the need of FEC at the ER levels we demonstrate in our work.

We can estimate the required ER based assuming reference systems that are either shot-noise limited (Scenario 1) or receiver noise limited (Scenario 2).

Scenario 1: Shot-noise limited system

One hypothetical scenario is to consider a shot-noise limited system, i.e, a system where the dominant source of noise is that associated with the power of the optical signal itself. In this case, the noise variance of the current at the detector is given by:

$$\sigma^2 = 2qIB$$

Where q is the electron charge, B is the signal bandwidth and I is the photocurrent generated by the input optical signal, $I = R * P_{rec}$. In the latter, R is the responsivity of the photodetector.

If we assume that, at the receiver site, we have a '1' current level I_1 and a '0' current level I_0 , we can write:

$$BER = \frac{1}{2} \operatorname{erfc} \left(\frac{I_1 - I_0}{\sqrt{2}\sigma_1} \right) + \frac{1}{2} \operatorname{erfc} \left(\frac{I_1 - I_0}{\sqrt{2}\sigma_0} \right)$$

Above, $\sigma_1 = \sqrt{2qI_1B}$ ($\sigma_0 = \sqrt{2qI_0B}$) is the shot noise when a bit '1' (bit '0') is received. If we use the fact that the ER is $ER [linear] = \frac{I_1}{I_0} = \frac{P_1}{P_0}$, substitute the expression for σ_1 and σ_0 and rearrange terms, we end up with the following expression:

$$BER = \frac{1}{2} \operatorname{erfc} \left(\sqrt{\frac{RP_1}{qB}} \frac{\left(1 - \frac{1}{ER}\right)}{2} \right) + \frac{1}{2} \operatorname{erfc} \left(\sqrt{\frac{RP_1ER}{qB}} \frac{\left(1 - \frac{1}{ER}\right)}{2} \right)$$

Above, P_1 is the optical power at the receiver site corresponding to the '1' bit. Given the expression above, it is easy to evaluate the achievable SNR (and from it the BER) for different ER, received powers and bandwidths.

The figure below shows the achievable BER as a function of ER for a 100 MHz bandwidth signal, assuming a photodetector with responsivity $R = 0.5 \text{ A/W}$, for different received powers:

Clearly, BER in the order of 0.5 dB are enough to achieve error-free communication (without the need for FEC) for received powers above 1 μW . For reference, our experimental results with a 7.5 μW input optical power and ≈ 3 dB IL correspond to $P_1 = 3.75 \mu\text{W}$.

Scenario 2: Receiver noise limited system

In this case, which is the most likely in practical receiver implementations, the electrical noise in the receiver dominates the total noise in the system.

In this case, the BER is given by:

$$BER = \frac{1}{2} \operatorname{erfc} \left(\frac{P_1 - P_0}{\sqrt{2} \sigma_{rec}} \right) = \frac{1}{2} \operatorname{erfc} \left(\frac{P_1 (1 - 1/ER)}{\sqrt{2} \sigma} \right)$$

Where σ_{rec} is the input equivalent current noise of the receiver circuit. We therefore need to obtain the input equivalent noise of the receiver to evaluate the SNR and the BER from it. Of course, that will depend on the implementation of the receiver, and can vary depending on the specifics of the receiver.

We can look for commercially available receivers to obtain some representative values of σ_{rec} so that we can evaluate BER.

1. Newport 125 MHz optical receivers (<https://www.newport.com/f/125-mhz-photoreceivers>):

According to the specifications, these have a noise equivalent power (NEP) = 2.5 pW/sqrt(Hz). If, as we did in the case of the shot-noise limited system, we plot the BER as a function of ER for different received '1' powers, we get the following:

Clearly, for received powers above $2 \mu\text{W}$, error-free communication can be established with $ER < 1 \text{ dB}$. IN the case of our experimental demonstrations, the received '1' power was $3.75 \mu\text{W}$.

Note: The NEP of the receiver analyzed here is representative of a great variety of commercially available optical receivers. For example, a 250 MHz receiver from Thorlabs (<https://www.thorlabs.us/thorproduct.cfm?partnumber=FPD510-FS-NIR>) has a $3.2 \text{ pW}/\sqrt{\text{Hz}}$.

2. Texas Instruments 125 MHz Transimpedance Amplifier (TIA) (https://www.ti.com/lit/ds/symlink/opa857.pdf?ts=1604249163803&ref_url=https%253A%252F%252Fwww.ti.com%252Fproduct%252FOPA857)

The datasheet specifies a total rms noise of 12 nA integrated over a 135 MHz bandwidth. Assuming a photodetector with responsivity $R = 0.5 \text{ A/W}$, the BER as a function of ER is the following:

Once again, for received powers above $2 \mu\text{W}$, error-free communication can be established with $ER < 1 \text{ dB}$.

Note 1: In this case, since we are only considering the TIA noise, we are assuming that the dark current of the photodetector is negligible.

Note 2: The noise power of the TIA analyzed here is representative of a great variety of commercially available TIAs. For example, Analog Devices has a TIA with 220 MHz bandwidth and 4.8 pA/sqrt(Hz) input current noise (<https://www.analog.com/en/products/ltc6561.html>).

In conclusion, while it is true that in conventional high-speed optical communications an ER of 1 dB is not enough for low error-rate communication, operating at low bandwidths enables error-free communications with ER on the order of 0.5 dB for received powers above $\approx 2 \mu\text{W}$, which we have demonstrated experimentally. A supplementary section (Supplementary Section S6) has been added reproducing (in a shorter format) the discussion here.

8. There is indeed a great deal of interest in minimizing the energy consumption of optical transmitters to fJ/bit levels or less. But this energy consumption specification includes the laser source and the driver electronics, which are often the dominant source of energy consumption and not the modulator itself. As the power dissipation of a laser is static, modulating at high bit rates has the advantage of reducing the energy per bit (= power / bit rate). In this article, the power consumption of the laser has been neglected. It would be instructive to quote the total energy per bit (including the laser source).

Response: The preliminary list of target applications (Fig. 1) require data rates in the Mbps range or below, and therefore there is no immediate benefit to having a modulator that could operate at much faster speeds: it is simply not needed.

In this case, it is more beneficial to have an optical modulator capable of operating with the minimal overall power dissipation (including laser power). This is what we discuss and quote in the last paragraph of the discussion section: the total power dissipation of $8 \mu\text{W}$ includes the laser power. Additionally, we have added a sentence at the end of the first paragraph of the discussion section quoting the optical energy dissipation per bit.

As discussed in response to point 2, even when accounting for the required optical power, the total power dissipation of our solution is lower than that of state-of-the-art solutions.

9. While this article has an interesting proposal, it seems to have missed the jugular when one focuses on energy consumption. I would recommend reframing the motivation of the article and presenting the work in a more balanced way that describes both the advantages and limitations of the work.

Response: As we have discussed in detail in our responses above, we believe that our photovoltaic modulator offers significant advantages for low power, low data rate applications compared to state-of-the-art solutions:

- When compared to electrical solutions, our scheme could provide significant gains in the total power dissipated for the readout of massively parallel data streams. In particular:
 - o For neural interface applications, the $7.5 \mu\text{W}$ total (electrical + optical) power and 7.5 pJ/bit energy dissipation demonstrated here represents a significant improvement over reported state-of-the-art electrical interfaces.

- Wired solutions have on the order of 100x higher power dissipation. For instance, the work in [Obaid et al., *Sci. Adv.* 6, eaay2789 (2020)] has 70 mW total power dissipation with a 20 kHz speed, which translates into 280 $\mu\text{W}/\text{channel}$ (14 nJ/bit per channel) for the demonstrated 250 channel array. Data available on the Neuralink probes [Musk et al., *J. Med. Internet Res.* 21, e16194 (2019)] reports a 5.2 μW analog power dissipation, which at the maximum operating bandwidth of 27 kHz translates into 190 pJ/bit energy dissipation.
- The lowest achievable energy dissipation in demonstrated wireless solutions are above 20 pJ/bit [de Marcellis et al., *IEEE Trans. Biomed. Circuits Syst.* 14, 441 (2020)].

Thus, our solution employing an optical link with a PV modulator could enable the readout of thousands of parallel sensors while still complying with the power dissipation limits set by the brain or any other biological tissue.

- For cryogenic readout applications, the use of optical links can significantly reduce the passive heat load of conventional electrical readout solutions due to the much lower heat conduction of optical fiber [de Cea et al., *Sci. Rep.* 10, 9470 (2020)]. Even when not accounting for the passive heat load of state-of-the-art electrical readout solutions, our demonstrated system has a competitive energy per bit dissipation.

Recently, novel approaches to cryogenic optical readout have been reported with energy dissipation in the 1 pJ/bit range [de Cea et al., *Sci. Rep.* 10, 9470 (2020), McCaughan et al., *Nature Electron.* 2, 451 (2019)]. Our proposed solution is still competitive with these, and does not have a high DC electrical power dissipation (unlike [de Cea et al., *Sci. Rep.* 10, 9470 (2020)]) nor requires the use of superconducting materials (unlike [McCaughan et al., *Nature Electron.* 2, 451 (2019)]).

- For MRI applications, power dissipation is not so critical. Instead, the ability to read out signals with improved robustness to electromagnetic interference is highly advantageous as it reduces the need for bulky and expensive electrical shielding.

The table included below summarizes the bandwidth and power dissipation of state-of-the-art approaches to the target applications described in our work. As can be seen, our proposed solution can provide significant gains in power dissipation per channel, which ultimately translates into an increase in the number of data streams that can be read out in parallel.

Target application	Ref.	Brief description	Bandwidth	Power dissipation	Energy per bit
Neural interfaces, cryogenic readout, MRI	This work	PV modulator	1 MHz demonstrated, 500 MHz achievable	7.5 μ W	7.5 pJ/bit dem, 0.1 pJ/bit achievable
Neural interface	[1]	Electrical neural interface, wired	27 kHz	5.2 μ W	190 pJ/bit
	[2]	Electrical neural interface, wired	20 kHz	280 μ W	14,000 pJ/bit
	[3]	Electrical neural interface, wired	30 kHz	25 – 50 μ W	830 – 1,660 pJ/bit
	[4]	Optical neural interface, wireless	300 MHz	11 mW	37 pJ/bit
	[5]	RF neural interface, wireless	136 MHz	3 mW	22 pJ/bit
Cryogenic readout	[6]	GaAs amplifier	1.5 GHz	3 mW *	2 pJ/bit
	[7]	nTron + HEMT	0.5 GHz	12 mW *	24 pJ/bit
	[8]	Optical modulator	1.5 GHz	1 mW	0.66 pJ/bit
	[9]	Superconducting thermal switch	100 MHz	100 μ W	1 pJ/bit

* Does not account for the passive heat load of the electrical connection between the cryogenic environment and the room temperature environment.

- [1] Musk et al., *J. Med. Internet Res.* 21, e16194 (2019).
 [2] Obaid et al., *Sci. Adv.* 6, eaay2789 (2020).
 [3] Jun et al., *Nature* 551, 232 (2017).
 [4] de Marcellis et al., *IEEE Trans. Biomed. Circuits Syst.* 14, 441 (2020).
 [5] Jung et al., *IEEE Trans. Microw. Theory Tech* 58, 4102 (2010).
 [6] Cahall et al., *Rev. Sci. Instrum.* 89, 063117 (2018).
 [7] Zhao et al., *Supercond. Sci. Technol.* 30, 044002 (2017).
 [8] de Cea et al., *Sci. Rep.* 10, 9470 (2020).
 [9] McCaughan et al., *Nature Electron.* 2, 451 (2019).

- Compared to state-of-the-art optical modulators, our device can achieve optical modulation with reduced electrical driving signals and reduced electrical power dissipation. While its speed is limited, it is enough for our low data rate target applications.

We also believe that by adding the optical energy dissipation per bit in the discussion section and modifying the abstract, the work is presented in a more balanced way.

REVIEWER COMMENTS

Reviewer #1 (Remarks to the Author):

Thank you for answering my questions and addressing the concerns. I have no further comments, and recommend that the paper move ahead with publication.

Reviewer #2 (Remarks to the Author):

The authors have tried to thoroughly address the comments raised in the first review. The revised manuscript, especially the introduction, is improved. Below are several comments that should be addressed before the manuscript can be accepted.

1. In Fig. 1(b) and introduction, it would be much clearer to state that electrophysiological signals are being detected. There are many types of biological signals. In (d), again, the device is meant to read out electrical or radiation signatures.

2. The BER discussion in the supplementary section needs revision and contains some errors.

a. First, the authors should state the BER they have considered to be error-free. In telecommunications, a BER of $1e-9$ is desired, but for data communications (like PCIeExpress, which seems to be targeted by the authors), a BER of $1e-12$ is required. I believe $1e-9$ is the criterion the authors used in the response letter, but it is not stated in the supplementary.

b. Second, the BER calculation should be revised. Normally, the BER is computed in the electrical domain (since a voltage or current is measured), so the variance (noise) in the denominator should be in the electrical domain (i.e., a current), and the numerator should be the detector current (let's call this I_1 and I_0) instead of the optical power.

Assuming the decision level is midway between I_1 (high) and I_0 (the "0" level), and the noise in the "1" and "0" levels are the same, the BER expression should be:

$$\text{BER} = 1/2 \operatorname{erfc}((I_1 - I_0)/(2 \sqrt{2} \sigma)).$$

Please note the factor of 2 in the denominator in the argument of the erfc function is missing in the authors' equation. For reference, please see a textbook, such as G. P. Agrawal, Fiber Optic Communication Systems, Wiley and Sons, Chapter 4.

By using optical powers in the BER calculations, the authors have therefore implicitly assumed an overall detection responsivity of $1A/W$, which may or may not be reasonable, depending on the system and photonic circuit. Nonetheless, if we assume a responsivity of $1A/W$ to convert the currents to optical powers and use the NEP of $25 \text{ pW}/\sqrt{\text{Hz}}$ with a 100MHz bandwidth to estimate σ , we find that for an ER of 1dB and $P_1 = 2\mu\text{W}$, the BER can still be less than $1e-9$ or $1e-12$. But for an ER of 0.5dB , the BER is $7e-6$, which is not low enough by conventional comm standards.

However, if we look at the data in the paper, the reality is very far from the calculations. For example, in Fig. 4(e) [the best case], we see that $I_1 \sim 630\text{mV}$, $I_0 \sim 550\text{mV}$ and can estimate σ to be around $10\text{mV}-12\text{mV}$. In this case, the computed BER is $3e-5$ to $4e-4$, which appears to be much too high.

A mention of the BER as extrapolated from the experimental results will be helpful and points to the need for future work. Revising the discussion around BER and the acceptability of the low ER is necessary.

Reviewer #1:

Thank you for answering my questions and addressing the concerns. I have no further comments, and recommend that the paper move ahead with publication.

Reviewer #2:

The authors have tried to thoroughly address the comments raised in the first review. The revised manuscript, especially the introduction, is improved. Below are several comments that should be addressed before the manuscript can be accepted.

1. In Fig. 1(b) and introduction, it would be much clearer to state that electrophysiological signals are being detected. There are many types of biological signals. In (d), again, the device is meant to read out electrical or radiation signatures.

Response: We have substituted “biological sensing” by “electrophysiological sensing” in the introduction (line 52). The caption in Fig. 1 has also been modified to state that Fig. 1(b) concerns the readout of electrophysiological signals, and that Fig. 1(d) is meant for readout of electrical signals in environments with high electromagnetic interference.

2. The BER discussion in the supplementary section needs revision and contains some errors.

a. First, the authors should state the BER they have considered to be error-free. In telecommunications, a BER of $1e-9$ is desired, but for data communications (like PCIeExpress, which seems to be targeted by the authors), a BER of $1e-12$ is required. I believe $1e-9$ is the criterion the authors used in the response letter, but it is not stated in the supplementary.

Response: We have extended our discussion to accommodate for both 10^{-9} and 10^{-12} BER requirements (see line 345 in the supplementary text). Additionally, the y axis in Supplementary Fig. S6 has been modified to show 10^{-9} and 10^{-12} BER.

b. Second, the BER calculation should be revised. Normally, the BER is computed in the electrical domain (since a voltage or current is measured), so the variance (noise) in the denominator should be in the electrical domain (i.e., a current), and the numerator should be the detector current (let’s call this I_1 and I_0) instead of the optical power.

Assuming the decision level is midway between I_1 (high) and I_0 (the “0” level), and the noise in the “1” and “0” levels are the same, the BER expression should be:

$$\text{BER} = 1/2 \operatorname{erfc}((I_1 - I_0)/(2 \sqrt{2} \sigma)).$$

Please note the factor of 2 in the denominator in the argument of the erfc function is missing in the authors' equation. For reference, please see a textbook, such as G. P. Agrawal, Fiber Optic Communication Systems, Wiley and Sons, Chapter 4.

By using optical powers in the BER calculations, the authors have therefore implicitly assumed an overall detection responsivity of $1A/W$, which may or may not be reasonable, depending on the system and photonic circuit. Nonetheless, if we assume a responsivity of $1A/W$ to convert the currents to optical powers and use the NEP of $25 \text{ pW}/\sqrt{\text{Hz}}$ with a 100MHz bandwidth to estimate σ , we find that for an ER of 1dB and $P_1 = 2\mu\text{W}$, the BER can still be less than $1e-9$ or $1e-12$. But for an ER of 0.5dB , the BER is $7e-6$, which is not low enough by conventional comm standards.

Response: The reviewer is right: there is a typo in supplementary equations S10 and S11, where there is a missing factor of 2. This has been corrected. The curves in Supplementary Fig. S6(a) have not changed, as the calculations were made with the correct formula.

When a vendor states a NEP number, the responsivity of the photodetector is already included (see for example eq. 4.4.14 in G. P. Agrawal, Fiber Optic Communications Systems, Wiley and Sons, Chapter 4). Therefore, we are not assuming a responsivity of $1 A/W$: supplementary equations S10 and S11 in the revised version are valid regardless of the photodetector responsivity.

However, if we look at the data in the paper, the reality is very far from the calculations. For example, in Fig. 4(e) [the best case], we see that $I_{-1} \sim 630\text{mV}$, $I_{-0} \sim 550\text{mV}$ and can estimate σ to be around $10\text{mV}-12\text{mV}$. In this case, the computed BER is $3e-5$ to $4e-4$, which appears to be much too high.

A mention of the BER as extrapolated from the experimental results will be helpful and points to the need for future work. Revising the discussion around BER and the acceptability of the low ER is necessary.

Response: We have added a paragraph at the end of Supplementary Section S6 (starting in line 407) discussing this:

“It is worth noting that the SNR achieved in the experimental waveforms shown in Fig. 4 in the main text are lower than the achievable SNR based on the calculations outlined above. This is due to the high optical losses associated to the coupling of light in and out of the chip. Because we use a non-optimized, broadband design, each grating coupler adds $\sim 10 \text{ dB}$ insertion loss to the optical path. Since the signal power decreases but the total noise (dominated by the receiver electrical noise) is not affected, this additional insertion loss translates into a decrease in SNR: a 10 dB loss output coupler results in a 10 dB decrease in our experimentally achieved SNR compared to what we could achieve if we were to use ideal couplers. For instance, based on the experimentally achieved SNR of the waveform in Fig. 4(e) (with an ER = 0.6 dB and $P_1 = 3.75 \mu\text{W}$)

we expect a BER $\approx 3 \cdot 10^{-9}$. If instead we use Eq. S11 for this ER and received power, a BER $< 10^{-13}$ could be realized.

This is a limitation of the chip design and the testing setup – not an intrinsic limitation of the device or the fabrication technology. We have demonstrated grating couplers with over 90% coupling efficiency (0.5 dB loss) in the same technology platform we use to fabricate our photovoltaic modulator devices (GlobalFoundries 45RF SOI) [23]. “

The noise standard deviation calculated from the waveform in Fig. 4(e) is $\sigma_1 \approx \sigma_0 = 7 \text{ mV}$.

REVIEWERS' COMMENTS

Reviewer #2 (Remarks to the Author):

Thanks for fixing the errors.